# Application of Nanoparticles in Cancer Treatment: A Concise Review

**DOI:** 10.3390/nano13212887

**Published:** 2023-10-31

**Authors:** Mariana Sell, Ana Rita Lopes, Maria Escudeiro, Bruno Esteves, Ana R. Monteiro, Tito Trindade, Luísa Cruz-Lopes

**Affiliations:** 1Polytechnic Institute of Viseu, Av. Cor. José Maria Vale de Andrade, 3504-510 Viseu, Portugal; pv23806@alunos.estgv.ipv.pt (M.S.); bruno@estgv.ipv.pt (B.E.); 2Faculty of Dental Medicine, Portuguese Catholic University, 3504-505 Viseu, Portugal; s-arvlopes@ucp.pt; 3Abel Salazar Biomedical Institute, University of Porto, 4050-313 Porto, Portugal; up201604443@up.pt; 4Centre for Natural Resources, Environment and Society-CERNAS-IPV Research Centre, Av. Cor. José Maria Vale de Andrade, 3504-510 Viseu, Portugal; 5Centro de Investigación en Química Biolóxica e Materiais Moleculares (CiQUS), Universidade de Santiago de Compostela, 15705 Santiago de Compostela, Spain; anarita.rvcm@ua.pt; 6Department of Chemistry, CICECO-Aveiro Institute of Materials, University of Aveiro, 3810-193 Aveiro, Portugal; tito@ua.pt

**Keywords:** cancer treatments, nanotechnology, nanoparticles, drug delivery, tumor environment, passive and active targeting, nanomedicine

## Abstract

Timely diagnosis and appropriate antitumoral treatments remain of utmost importance, since cancer remains a leading cause of death worldwide. Within this context, nanotechnology offers specific benefits in terms of cancer therapy by reducing its adverse effects and guiding drugs to selectively target cancer cells. In this comprehensive review, we have summarized the most relevant novel outcomes in the range of 2010–2023, covering the design and application of nanosystems for cancer therapy. We have established the general requirements for nanoparticles to be used in drug delivery and strategies for their uptake in tumor microenvironment and vasculature, including the reticuloendothelial system uptake and surface functionalization with protein corona. After a brief review of the classes of nanovectors, we have covered different classes of nanoparticles used in cancer therapies. First, the advances in the encapsulation of drugs (such as paclitaxel and fisetin) into nanoliposomes and nanoemulsions are described, as well as their relevance in current clinical trials. Then, polymeric nanoparticles are presented, namely the ones comprising poly lactic-co-glycolic acid, polyethylene glycol (and PEG dilemma) and dendrimers. The relevance of quantum dots in bioimaging is also covered, namely the systems with zinc sulfide and indium phosphide. Afterwards, we have reviewed gold nanoparticles (spheres and anisotropic) and their application in plasmon-induced photothermal therapy. The clinical relevance of iron oxide nanoparticles, such as magnetite and maghemite, has been analyzed in different fields, namely for magnetic resonance imaging, immunotherapy, hyperthermia, and drug delivery. Lastly, we have covered the recent advances in the systems using carbon nanomaterials, namely graphene oxide, carbon nanotubes, fullerenes, and carbon dots. Finally, we have compared the strategies of passive and active targeting of nanoparticles and their relevance in cancer theranostics. This review aims to be a (nano)mark on the ongoing journey towards realizing the remarkable potential of different nanoparticles in the realm of cancer therapeutics.

## 1. Introduction

Cancer is one of the leading causes of death in the world, as shown by the 2022 cancer statistics, predicting 1,918,030 new cases of cancer and 609,360 related deaths per year [1].

The classic therapeutic options when approaching a cancer patient are chemotherapy, radiotherapy, and surgery. The choice of approach depends on several characteristics, such as the cancer stage and location or patient’s fitness, which is compromised by the disease itself, worsening with each treatment intervention in the long term [2]. These treatments can reduce cancer recurrence and mortality but have important side effects that can lead to severe complications and to the risk of death from other diseases [3].

Radiotherapy has been for a long time an extremely important tool against cancer, offering a possible cure, symptoms relief, and extended survival. Nonetheless, it is linked to important side effects. When a patient is exposed to radiotherapy, not only are the tumoral cells targeted but the normal tissue around the tumor is also damaged [4].

As for chemotherapy, many pharmacological classes can be used in the treatment of cancer, showing potential side effects such as autoimmune-like disorders and fatal adverse events caused by the reactivation of cellular immunity [5].

There have been great efforts in diverse scientific fields to limit the abovementioned problems by exploring alternatives which prevent the toxicity and side effects associated with conventional therapies. Overall, most of these new approaches are still the object of intense research, such as the exploitation of surface-modified inorganic nanoparticles to fight cancer [6,7,8].

However, they have proved to face substantial limitations. The main disadvantages of chemotherapy and radiotherapy have been the lack of specificity, which causes the drug delivery in the targeted site to be of inadequate concentrations, and also high toxicity to the healthy and surrounding cells, tissues, and organs, leading to the development of drug resistance during the treatment [9]. The scientific community leans on the use of nanotechnology, as a strategy with great potential to overcome these challenges [10], specifically by enhancing the drug delivery into target sites, increasing efficacy and reducing side effects [11]. In this regard, the significant specific surface areas of nanoparticles give them useful properties, such as the ability for biofunctionalization and a valuable interface to mediate processes involving the nanoparticles and the surrounding tissues [12]. Nowadays, a variety of products involving the synthesis of nanoparticles or their use are being developed and nanomedicine is becoming an attractive research field thanks to its potential efficacy and requirement of smaller amounts of drugs [13]. Therefore, the use of nanoparticles in this context might also contribute to enhancing, stimulating or improving the effectiveness of drug treatment at more affordable costs [14,15]. Nanoparticles have ushered in a paradigm shift in the domain of drug delivery for oncological interventions. Specifically, liposomes, polymeric nanoparticles, and lipid nanoparticles have been meticulously engineered to encapsulate and facilitate the conveyance of chemotherapeutic agents. In doing so, they have adeptly surmounted hurdles associated with drug solubility and systemic toxicity, heralding the emergence of multiple nanoparticle-based drug delivery systems currently traversing various stages of clinical development [16,17,18].

The amalgamation of imaging methodologies into the realm of cancer diagnostics and therapeutic monitoring has been significantly amplified through the strategic integration of nanoparticles. These diminutive structures, when judiciously loaded with imaging moieties such as quantum dots and gold nanoparticles, furnish the capability for instantaneous visualization of neoplastic lesions and the real-time tracking of therapeutic agent dissemination [19,20].

The burgeoning frontier of “theranostics”, an innovative concept fusing therapeutic and diagnostic functionalities, has garnered substantial attention in the realm of nanoparticle research. Certain nanoparticles have been ingeniously tailored to serve a dual purpose by concurrently acting as drug carriers and bestowing invaluable imaging capabilities. This dual functionality augments the precision and efficacy of cancer treatments, currently under extensive exploration in the clinical trial landscape [14,21].

Magnetic nanoparticles have been adroitly harnessed in the domain of hyperthermia therapy, leveraging the application of alternating magnetic fields to selectively induce controlled hyperthermic effects within cancer cells. Clinical trials have been conducted to assess the feasibility and therapeutic potential of this methodology, particularly for distinct malignancies [22,23].

Gold nanoparticles, along with their counterparts, have demonstrated remarkable promise in augmenting the sensitivity of cancer cells to radiation therapy. These nanoparticles, when judiciously targeted to neoplastic foci, potentiate the absorption of radiation energy, thereby intensifying cellular damage. The ongoing clinical research is devoted to elucidating the practical utility and therapeutic effectiveness of this strategy in the context of radiation-based cancer treatment [19,24,25].

In the realm of precision oncology, nanoparticles offer the tantalizing prospect of delivering treatments with unparalleled specificity. Through the functionalization of nanoparticles with ligands tailored to selectively home in on cancer cells, the collateral damage to healthy tissues is mitigated, and the overall efficacy of therapeutic interventions is substantially augmented [26].

Herein, we briefly review the application of fine divided systems termed nanoparticles, which are characterized by dimensions typically from 1–100 nm and show properties strongly dependent on size and surface [27,28]. As such, non-conventional particulate systems that have been explored for cancer therapies are mostly surface-functionalized inorganic nanomaterials, typically obtained as stable colloidal nanoparticles [6,25,29]. However, other types of nanoparticles have been investigated for a longer time in the context of cancer therapies, such as diverse polymeric nanoparticles and nanoliposomes; the latter are well-known drug carriers [16,30].

Considering the diverse types of colloidal nanoparticles available, their systematic classification is challenging, and several attempts have been implemented. Hence, nanoparticles can be classified according to their shape, average size, chemical nature, and preparation method, among other criteria [31,32]. The classification of nanomaterials proposed by Miernicki et al. goes further in the parametric assessment by considering also their applicability and safety, besides the morphological characteristics [33]. A number of applications mediated by surface phenomena take advantage of the high ratios of surface area per volume that characterize nanoparticles. For example, specific surface area and surface functionalization are important aspects to take into consideration in the application of nanoparticles for drug delivery. An increased surface area available implies an increased amount of anticancer agents that can be attached, making them more efficient candidates for drug delivery vectors [9]. Due to their nanometric size, nanoparticles are able to cross pores which contributes to more effective treatments for neurological conditions and brain cancer due to their ability to go through the blood–brain barrier [34,35].

According to Fraga et al. [36], there are many abilities in favor of nanosized therapeutic development and one of them is the nanoparticle’s ability to overcome solubility and stability problems of anticancer drugs. Since the bioavailability is restricted by water solubility and it hampers the development of early agents of drugs, the delivery and consumption of a poorly soluble drug can be increased by an encapsulation of the compound within a hydrophilic nanocarrier [13]. Another way of protecting anticancer compounds from excretion or decomposition requires the encapsulation of antineoplastic agents in nanocarriers or pairing perishable compounds with synthetic ones [13]. Also, nanotechnology can improve drug penetration and redirection or selectively redirect compounds to cancer cells through its physicochemical properties. For redirection of antitumor drugs, active and passive targeting schemes are used.

Additionally, nanocarriers are made to expel their cargo at the beginning, so it results in a stimuli-sensitive nanomedicine treatment. For example, a medicine that is pH-independent, like doxorubicin, can be catenated to pH-sensitive nanoparticles to increase the cellular uptake and intracellular release of the medicine [37]. Eventually, the endurance of tumors is attenuated against antitumor medicines through guided nanomedicine treatments. Generally, non-specificity is decreased by targeted input and multidrug resistance/adenosine triphosphate outflow pump-driven excretion. Therefore, the circulation time of a drug can be held by nanomedicine, helping the release of stimuli-responsive medicines to intervene in endocytic input of the drug [13].

## 2. Methodology

This article was written based on a bibliographic survey that was performed by consulting relevant online scientific communication platforms like ScienceDirect, Web of Science, Taylor & Francis, and Scielo. A search was made on those platforms to find the major contents related to the theme of this review: the advantages and disadvantages of using nanotechnology in cancer treatments, the nanoparticles’ toxicity, the innovations in the treatment of cancer, and the concerns about unknown potential long-term effects.

In this article, we have covered a wide array of references spanning a time frame from 1994 to 2023, with a greater focus on the period between 2014 and 2023 (110 out of 188 references). Figure 1 illustrates the correlation between the number of publications in the field of this review and the temporal dimension, specifically focusing on the last ten years.

The main keywords used to carry out the bibliographic research were: “nanotechnology”, “cancer treatment”, “drug delivery”, “nanoparticles”, and “nanomedicine”. Figure 2 shows a cloud obtained from the keywords that appeared in all the sources that were used in this research, and which occurred at least five times in different articles. It can be observed that the most frequent keywords used were “nanoparticles”, “cancer”, “cancer treatment”, and “drug delivery”, which is in accordance with the main subject of this review.

The analysis with the VOSviewer software selected 15 keywords, grouped in three clusters with 66 links and a total link strength of 160. In Table 1, the clusters are shown and the score for each one is calculated as the average publication year of the documents in which a keyword or a term occurs.

The literature review focused on the critical analysis and presentation of information related to the application of nanotechnology in cancer treatments, from diagnosis and imaging to the mechanisms of drug delivery. Furthermore, a selected bibliography was also considered for supporting aspects that have been reviewed in the abovementioned context.

## 3. Tumor Microenvironment and Vasculature

It is known that the tumor microenvironment acts as a barrier to avoid drug delivery due to its poor vasculature, high interstitial fluid pressure, and dense extracellular matrix. Therefore, it is important to understand the tumor’s structure and specific aspects to reach an efficient drug delivery able to fight it [38].

The vasculature of a tumor is characterized by extra production of angiogenic factors which means that new blood vessels originate from already-existing blood vessel structures, resulting in convoluted and leaky vessels (Figure 3) [39]. This process can come as an advantage but also shows limitations for nanoparticle drug delivery. The growth of new vascular vessels increases the enhanced permeability and retention (EPR) effect which allows nanoparticles to be discharged from vessels and accumulate inside the tumor [40]. Nevertheless, it can also happen with blood components, blocking the overflow of nanoparticles. Furthermore, some areas inside the tumor have a lack of perfusion which creates an acidic and hypoxic environment and leads to the advancement of the tumor, increasing its resistance, and making the drug delivery more difficult [38,41].

The high interstitial fluid pressure, which is caused by abnormal blood flow and impaired venous and lymphatic drainage [42], is the reason why the extracellular matrix of a tumor is so dense. Similarly to vasculature, it can increase the EPR effect of nanoparticles, but it can also result in the limitation of fluid transfer, blocking its penetration through the tumor. Solid stress, which means the disorderly proliferation of tumor cells, is another reason that hinders the drug delivery: it weakens the immune response as it expands the cancer cells’ invasion [38].

## 4. General Requirements for Nanoparticles in Drug Delivery

Nanoparticles must have some properties to attain an effective system for cancer treatment such as being biocompatible, of high bioavailability, and stable under physiological conditions. Furthermore, they must be able to target only the tumor cells without deteriorating surrounding healthy cells and need to release the load as soon as they reach the target site. All these features can be affected by the physicochemical properties of the nanoparticles employed as drug delivery vectors (Figure 4) [43,44].

The nanoparticles’ size distribution has a major effect on their performance in cancer therapies. Due to the tumor’s leaky vasculature, the size of nanoparticles can be adapted to be small enough to penetrate the tumor and big enough to prevent extravasation from normal blood vessels, preventing agglomeration in other parts of the body [44]. Nevertheless, different organs have different size uptake specifications. Therefore, many types of research have been carried out to define the adequate size of nanoparticles used in cancer therapy, showing that smaller-sized particles (<50 nm) present better antitumoral efficiency rates than larger-sized particles [46].

Another important feature when it comes to nanoparticle design for cancer treatment is its shape, since it influences fluid dynamics, among other effects [40]. The shape of a nanocarrier can control the interaction between cell membrane and nanoparticles. It is also noted that the particle’s shape influences whether the nanoparticles are taken up by the reticuloendothelial system (RES) [40]. In fact, the accumulation of nanomedicines in by the RES organs (namely the liver) remains as a major hurdle to their clinical translation, since the RE cells retain the majority of the injected dose (decreasing the bioavailability of the injected nanoparticles), thereby increasing the associated immunogenicity and toxicity [17,47]. A possible strategy to overcome this problem is to promote the temporary blocking of the RE cells, increasing drug delivery efficiency to the disease sites by rerouting the nanoparticles from the RE organs without the need for special ligands [48,49,50,51].

Surface chemistry comprises a rather complex set of processes that includes, namely, surface charge, porosity, defects, and chemical group alterations. Numerous system features, such as surface interactions, degradation and agglomeration rates, and cellular uptake, are influenced by surface chemistry. Some studies imply, for instance, that a positively charged surface raises the chances of cellular uptake [43,52]. However, different types of cancer or different stages of the same cancer type can require different surface properties [40], as shown in Figure 5.

In biological environments, the surface of a nanoparticle is rapidly coated with a layer of biomolecules, which due to its high protein content is commonly known as “protein corona” (PC) [53,54,55,56]. The characteristics of PC (such as molecular properties and composition) play a key role in governing the cellular uptake, biocompatibility, distribution, and circulation lifetime of nanoparticles [54,57]. Two of the major factors influencing protein adsorption are the structural stability of proteins (namely their conformational changes) and the hydrophilic/hydrophobic properties of the surface of nanoparticles. With the dynamic behavior of proteins in physiological media, the layers of PC could be divided into hard corona (inner layer, with tightly bound proteins) and soft corona (weakly bound proteins, rapidly exchanged with free proteins from the media), as illustrated in Figure 6 [58]. Several techniques have been employed to separate, identify, and quantify the composition of PC’s layers, such as dialysis, centrifugation, gel filtration, size-exclusion chromatography, and high-pressure liquid chromatography, coupled to spectroscopic methods and bioinformatics predictions [59,60]. An in-depth understanding of the corona-mediated functionalities can be explored for the development of anticancer strategies. For instance, five distinct types of human cancers (lung, glioblastoma, meningioma, myeloma, and pancreatic cancers) have already been identified and discriminated by Caracciolo et al. by using liposome-based nanoparticles with three distinct surface properties; the authors developed a successful detection platform based on a PC sensor array [61]. Still, it is worth mentioning that PC might also negatively affect the delivery fate of nanoparticles for tumor targeting, particularly if their main components are dysopsonins [62] or if the components of PC promote an unfavorable steric effect that hampers the interactions with cell membranes (and, consequently, decreases cellular uptake) [63,64,65]. As so, due to the high complexity and heterogeneity of PC, future research is required in this field to allow the development of safe and effective nanoparticle-based applications.

## 5. Nanoparticles in Cancer Therapies and Clinical Diagnosis

Recently, many accomplishments have been achieved in the field of nanomedicine regarding drug delivery systems. Among them, an abundant number of nanoparticle types have been developed to be used in cancer therapy due to their unique properties [66], as described throughout this section and exemplified in Table 2.

A nanovector is generally defined as a functionalized nanoparticle that can carry and deliver anticancer drugs or detection agents. Nanovectors have been classified into three different classes: first-, second-, and third-generation systems (Figure 7) [10].

As an example of a first-generation nanovector, there is albumin-bound paclitaxel [79]. Paclitaxel can be used in breast cancer treatments and its solubility problem is solved by using Cremophor EL. However, first-generation nanovectors are not able to target any specific biomolecule in a tumor cell [10].

The second generation is an evolution of first-generation nanovectors and these are able to target a specific biomolecule in a tumor cell, which means they have active targeting capability. Examples of nanovectors from this generation are the antibody-targeted nanoparticles [79], such as mAb-conjugated liposomes [10]. The nanovectors of the second generation have an improved biodistribution and present a reduced toxicity level when compared to the first generation [10].

The third-generation nanovectors, such as the nanoshuttle, are multistage agents and can handle more complex functions [79]. According to Chatterjee and Kumar, this generation represents the next generation of the first-wave nanotherapeutics that are specially equipped to introduce biological barriers to improve the drug delivery to the tumor site [10].

The subsections below provide a summary of important nanoparticles that have been used in drug delivery and other clinical applications for fighting cancer. There are no attempts to provide a detailed description of the selected nanoparticles but rather an indication of their potential in the context approached in this review.

### 5.1. Nanoliposomes and Nanoemulsions

Liposomes, which are made up of non-toxic and biocompatible lipid bilayers, can act as pharmaceutical carriers [16]. Their core is aqueous, their head is hydrophilic, and the tails are hydrophobic, which means they are oriented away from the intercellular fluid. The conventional nanoparticle size is up to 100 nm and liposomes fluctuate between 90 and 150 nm. Liposomes are used to deliver the drug to the outer membrane of targeted tumor cells and, meanwhile, the fatty layer protects the enclosed drug [80]. This mechanism can decrease the effect of drug toxicity on healthy cells and increase the efficacy [10].

Liposomes can be synthesized from cholesterol and phospholipids and they have one particular property, which is their amphipathic nature, that enables them to bind to both hydrophilic and hydrophobic compounds [40]. In other words, they can encapsulate water-soluble drugs in their core and non-polar compounds in their bilayer membrane simultaneously [40]. Liposomes have other advantageous properties, like biocompatibility and biodegradability, and they do not present toxicity or immunogenicity [18,81]. Moreover, the Food and Drug Administration (FDA) has already approved drug delivery systems based on liposomes like MyocetTM [82].

Zhang et al. developed a lyophilized system based on liposomes and paclitaxel applicable for cancer therapy [14]. It has an encapsulation efficiency of over 90% and physical and chemical stability for 12 months while the particles have an average size of about 150 nm. When the system was diluted, the size remained the same and the drug was encapsulated.

Zhao et al. focused on a pH-responsive liposome-containing system for glioma tumor cells [83]. As the system is made up of a tumor-specific pH-sensitive peptide and liposomes, it responds to the acidic pH of gliomas and releases the drug. The same occurs when doxorubicin is used.

Theranostic systems based on liposomes have been studied to be used in imaging and drug delivery, as shown in Figure 8 [10]. Ren et al. designed a system in which a pharmaceutically active component was encapsulated and its biodistribution was imaged in real time by magnetic resonance imaging (MRI) [84]. The system was compared to a commercially available MRI contrast agent called Omniscan^®^ and showed not only better results but also a longer circulation time in vivo. Furthermore, liposomes enable the entrapment of both polar and non-polar chemotherapeutic drugs providing synergetic therapy with sustained release and substantially lower toxicity.

In recent years, nanoemulsions have also been attracting the interest of researchers in cancer therapies [85,86,87], due to their advantageous characteristics when compared to nanoliposomes, such as larger surface area, elevated half-life circulation, specific targeting, superficial charge, and imaging capacity. Nanoemulsions are heterogeneous emulsions (droplet size of ~100 nm) that simultaneously contain oil, water, and an amphiphilic emulsifier. The dosage form can be tuned to optimize the stability and solubility of drugs in different environments: in this way, poorly water-soluble drugs can be encapsulated into nanoemulsions with a hydrophobic nature, protecting them from degradation and increasing their half-life in the plasma [88].

For example, Ragelle et al. [67] described that the incorporation of fisetin (a naturally occurring flavonoid) into nanoemulsions enhances its water solubility, bioavailability, and efficacy. The formulation was composed of Miglyol^®^ 812N/Labrasol^®^/Tween^®^ 80/Lipoid E80^®^/water (10%/10%/2.5%/1.2%/76.3%), with a droplet diameter of ~153 nm. When administered intraperitoneally, the nanoemulsion showed a 24-fold increase in fisetin relative bioavailability when compared to free fisetin. Therefore, the antitumoral activity in mice bearing Lewis lung carcinoma was improved in the nanoemulsion (36.6 mg/kg) when compared to free fisetin (223 mg/kg).

Hu et al. developed a nanoemulsion formulation containing an oil phase (oil with lycopene), water phase (aqueous gold nanoparticle solution), and an emulsifier (Tween 80^®^), which showed promising results in the regression of a human colon cancer cell line (HT-29) [68]. Briefly, the authors found out that this formulation decreased the expression of procaspases 3 and 8 and Bcl-2 (tumoral markers), while enhancing Bax and PARP-1 expression, accompanied by apoptotic cell death.

In another study, Kretzer et al. developed lipid nanoemulsions containing paclitaxel, which were able to bind to low-density lipoprotein receptors, thus decreasing the drug toxicity and antitumoral potential [89].

Nanoemulsions are currently being used for clinical trials, such as superficial basal cancer cell photodynamic therapy (ClinicalTrials.gov ID: NCT02367547) [90], treatment of lentigo maligna (ClinicalTrials.gov ID: NCT02685592), multiple actinic keratosis (ClinicalTrials.gov ID: NCT01893203), and actinic keratosis (ClinicalTrials.gov ID: NCT01966120 and NCT02799069). However, up to the present, no formulation of this type has been approved by the FDA, as the long-term stability and safety have yet to be further studied.

### 5.2. Polymeric Nanoparticles

Polymeric nanoparticles have been considered efficient carriers for prolonged drug delivery systems. In the 1990s, the synthesis of polymeric nanoparticles using polylactic acid (PLA) and poly lactic-co-glycolic acid (PLGA) was explored and reported as “long-circulating” [91]. Since then, the interest in polymeric nanoparticles and their use in cancer therapy has increased. These nanoparticles are considered very versatile because they can be manipulated to be either biodegradable or non-biodegradable, either synthetic or derived from natural sources [92,93]. Biodegradable polymers have the advantage that they can break down into monomers that can be simply eliminated by the body’s natural metabolic pathways [40].

Natural polymers such as polyhydroxyalkanoates (PHAs), as well as synthetic polymers like PLGA, have been studied for targeted drug delivery applications paired with anticancer agents like paclitaxel [70,94], doxorubicin [95,96], and cisplatin [97,98]. These studies were tested in vivo and there are some that have been used in preclinical trials on mice [99].

A conjugation between folic acid and PLGA nanoparticles with chitosan as the vehicle was tested for the treatment of prostate cancer [71]. The compound was loaded with bicalutamide and tested in vitro. In comparative studies, unfunctionalized PLGA nanoparticles were also synthetized and exposed to the same circumstances. It was observed that the functionalized nanoparticles showed improved efficiency compared to the unfunctionalized nanoparticles, because of their altered surface and specific targeted delivery [71]. Folic acid coupled with poly(3-hydroxybutyrate-co-3-hydroxyoctanoate) and loaded with doxorubicin presented a drug encapsulation performance of above 80% [96].

Furthermore, in vitro assays exhibited a release profile of the anticancer drug of approximately 50% in the first five days, and in vivo assays showed that the system displayed enhanced therapeutic efficiency in limiting the tumor growth when compared to controls [96]. Additionally, PLGA nanoparticles loaded with methotrexate–transferrin conjugates and coated with Polysorbate 80, a water-soluble surfactant, were investigated as vehicles for brain cancer treatment. Polysorbate 80 is known to enhance the transport of nanoparticles across the blood–brain barrier (BBB) [100,101]. According to Jain and al., the continuous delivery of methotrexate–transferrin conjugates was attained by virtue of the overexpressed transferrin receptors on the surface of tumor cells, and the results of both in vivo and in vitro assays highlighted the efficiency of the conjugated system when compared to controls [102].

The surface functionalization of nanoparticles with polyethyleneglycol (PEG), also termed PEGylation, is a widely used strategy for extending their blood circulation, thereby improving therapeutic outcomes in vivo. However, PEGylation compromises the uptake and endosomal escape efficiency (PEG dilemma). To overcome this dilemma, several strategies were introduced regarding the surface of nanoparticles to improve cancer treatment and diagnosis. For example: a polyion complex micelle was developed by self-assembling ethylenediamine-based polycarboxybetaine polymers with pDNA [103]. This micelle switched its surface charge to a positive charge in response to a tumorous (pH 6.5) and endolysosomal acidic milieu (pH 5.5) from its original neutral charge at pH 7.4 (bloodstream), thereby promoting the cellular uptake and endosomal escape toward efficient gene transfection. The cargo pDNA of this micelle encodes a soluble form of soluble fms-like tyrosine kinase-1, a potent antiangiogenic exogenous protein, which captures vascular endothelial growth factor (VEGF), thereby significantly suppressing the growth of hard-to-treat solid tumors.

In another example, ligands targeting tumor neovasculature endothelial cells (for example, cyclic Arg-Gly-Asp) are strategically appended to the distal end of the PEG shell for promoting tumor cell uptake of nanoparticles via specific integrin-mediated uptake [104].

Recent results challenge the transport of nanoparticles through interendothelial gaps of the tumor blood vessels, which is a central paradigm in cancer nanomedicine. Sindhwani et al. found that up to 97% of nanoparticles enter tumors using an active process through endothelial cells, unlocking strategies to enhance tumor accumulation [105,106].

Additionally, among the polymeric nanoparticles, dendrimers stand as a unique class of macromolecules with narrow molecular weight distribution, comprising an almost monodispersed nanosystem for target drug delivery. They are composed of a hyperbranched polymeric mantle, a central core, and corona and have numerous branches that can carry a variety of drugs [107]. The molecular size of dendrimers of a certain family is very often identified by its generation, which increases as the molecular weight of the dendrimer increases. The particle size and shape of dendrimers can be adjusted via chemical synthesis, thus providing branched macromolecules with diverse chemical groups that can be explored for target applications. This is of uttermost relevance for drug delivery because the loading of guest species (e.g., drug molecules) depends on the nature and number of chemical groups in the branched architecture. Due to their single surface, dendrimers have made a great contribution to the design of nanosystems but cytotoxicity has been a critical issue in these systems; the toxicity of these nanocarriers has been related, namely, to surface terminal groups [108]. The most valuable ability of these nanoparticles is the active and passive tumor targeting.

### 5.3. Quantum Dots

Quantum dots are semiconducting nanocrystals whose charge carriers are confined in the three dimensions, thus showing quantum size effects in their optical properties [27,28]. These inorganic nanoparticles have been prepared by a variety of chemical methods, however, those relying on colloidal synthesis offer several advantages for nanomedicines such as their easy biofunctionalization, namely, for bioimaging diagnosis. Among the biomarkers used for these purposes, quantum dots stand out for their size-dependent photoluminescence, narrow and tunable emission bands, photostability, and pronounced Stokes shift. Furthermore, the observation of size-tuned photoluminescence in quantum dots under irradiation using a single light source makes these particles suitable for multiplexing methods of analysis. Seminal research on colloidal quantum dots involved mainly the synthesis of Cd-containing materials using hot injection methods, whose surfaces could be subsequently modified with biomolecules. Currently, alternatives to toxic Cd-containing quantum dots are available and have been a subject of interest for bioimaging, such as zinc-sulfide-coated indium phosphide quantum dots or other types of fluorescent nanoparticles, including silica nanocomposites [109,110].

The present imaging techniques available such as X-ray scan, MRI, and computer tomography have serious limitations when it comes to cancer diagnosis and the main limitation is that those techniques cannot recognize small numbers of malignant cells in primary or in metastatic sites [111]. Because quantum dots have improved signal brightness, synchronous excitation of multiple fluorescence colors, and size-tunable light emission, they have been explored as biofunctionalized labels for cancer imaging [10]. However, besides the requirements for cytotoxicity assessment, quantum dots still pose challenges concerning their use in bioimaging, such as the observation of tissue autofluorescence and photon scattering.

### 5.4. Gold Nanoparticles

Throughout history, gold has consistently held its place as one of the most prized metals on Earth. Gold nanoparticles have been widely investigated for cancer therapies, due to their high chemical stability, well-established synthetic and surface modification methods, shape and size tunability, and biocompatibility [20,24,112,113]. In addition, gold nanoparticles show strong absorption in the visible spectrum due to localized surface plasmon resonances (LSPRs); this means that in the presence of light (an oscillating electromagnetic field), the free electrons from these plasmonic nanoparticles will oscillate and resonate at a particular frequency of light [114]. In fact, gold nanospheres (Figure 9A) absorb light up to 10^5^ times stronger than most efficient light-absorbing dye molecules [115], which is a clear advantage in comparison to the conventional drugs. The LSPR oscillation can decay by non-radiative processes and convert energy to heat, which makes gold nanoparticles particularly important for plasmonic photothermal therapy (PPTT) applications [116]. Furthermore, anisotropic gold nanostructures (e.g., gold nanorods or gold nanostars, Figure 9B) can be synthesized to show resonances in the near-infrared windows (650–950 nm; 1000–1700 nm), a spectral range that allows maximum depth of penetration of incident light in a tissue. Hence, a gold nanorod shows two LSPR bands, associated with two dipole oscillations along its axis, the transverse and longitudinal modes. The latter originates strong absorption in the NIR spectrum, whose exact location can be adjusted by controlling the particle’s aspect ratio during the synthesis. The ability for controlling the plasmonic behavior of gold nanoparticles via chemical and surface modification methods makes these nanosystems of great relevance in a number of cancer therapies, including PPTT and surface-enhanced Raman scattering bioimaging [25,29,117,118,119,120].

Considering the capability to thermally destroy the cancerous cells, the photothermal heating capacity, and ease in surface functionalization, gold nanoparticles stand out for their application in multiple cancer therapies. According to Lungu et al., hyperthermia, a common approach in terms of cancer treatment, consists in heating the tumor site up to 40 °C using microwaves and radiowaves as heat generators [40]. Nonetheless, gold nanoparticles can be used as heat sources, showing many advantages over conventional hyperthermia, such as the ability to affect only the adjacent sites, without damaging healthy cells, leading to efficient targeted action [121,122]. The gold nanoparticles start heating up the adjacent locations when an external radiofrequency electric field acts upon them. However, radiofrequency hyperthermia has some serious inconveniences, such as high levels of pain for the patient [121].

It has been reported that gold nanoparticles generate local dose augmentation at the cancerous location by virtue of their properties, such as strong optical absorption in the LSPR region. Furthermore, a system consisting of gold nanoparticles and organic molecules, such as bovine serum albumin (BSA), results in a higher agglomeration of nanoparticles in the tumor site [40]. Also, this system exhibits better features, such as uniform dimensions, ease in synthesis, and stability under physiological conditions [40]. According to Chen et al., both in vitro and in vivo assays using BSA-modified gold nanoparticles showed auspicious results, such as inhibition of cloning formation and cancerous cell death and did not present destructive consequences on healthy tissues and cells [108].

The basis of using gold nanoparticles in cancer radiotherapy is to inject them into the tumor location, then the external X-ray source will act upon them, and it will produce radicals that will damage the cancerous cells and promote their death [40]. When it comes to radiotherapy, assays were made by injecting gold nanoparticles in mice with the EMT-6 cancerous cell line. The mice were exposed to X-ray therapy and the survival rate considerably increased compared to mice that were subjected to conventional treatment, such as irradiation [40].

As mentioned above, colloidal Au nanoparticles can be synthesized with distinct particle size distributions and specific particle shape, such as nanospheres and anisotropic particles (Figure 9). As such, the optical behavior of such colloids can be judiciously tuned by controlling their morphological characteristics. Additionally, the surfaces of such nanoparticles can be functionalized envisaging specific bioapplications. Hence, Au nanoparticles coated with cysteamine and thioglucose were synthesized by Kong et al. and applied to healthy and cancerous breast cell lines [123]. It was reported that the gold nanoparticles coated with glucose were internalized by the tumor cells, while the ones coated with cysteamine were essentially disposed on the surface. The assays showed that the number of internalized functionalized nanoparticles was substantially higher than that of the unfunctionalized ones. Nonetheless, when the irradiation acted upon the nanoparticles, it was noticed that the cytotoxic effect of the functionalized nanoparticles was considerably higher than the one arising from the unfunctionalized ones.

### 5.5. Iron Oxide Nanoparticles

Iron oxide nanoparticles, namely of magnetite (Fe_3_O_4_) and maghemite (γ-Fe_2_O_3_), have garnered significant attention in cancer therapies (Figure 10), due to their unique properties, such as small size, high surface-to-volume ratio, and magnetic properties (which differ from their bulk counterparts) [124,125]. Often, the surface of these magnetic nanoparticles is coated with a material that increases the biocompatibility and stability in physiological media, such as a polysaccharide or smaller carbohydrates, endowing the final material with a hard-core/soft-shell structure [126].

One of the primary applications of these nanoparticles is their use as contrast agents for MRI scans (such as ferumoxsil, Lumirem^®^, or Gastro MARK^®^), enabling accurate tumor localization, staging, and monitoring of treatment response [75]. For example, Han et al. developed multifunctional iron oxide nanoparticles with a carbon-based shell, whose magnetic and fluorescence properties allowed the detection and imaging of cancer cells [127]. To provide an optimal balance of sensitivity and selectivity, MRI-based approaches can be combined with other imaging techniques, such as computerized tomography, which relies on the application of X-rays to generate two-dimensional images of the body. Within this context, Deng et al. [128] reported the synthesis of radiolabeled superparamagnetic iron oxide nanoparticles functionalized with a small peptide, as selective dual-modality agents for imaging of breast cancer.

Another well-documented application is the use of magnetic iron oxide nanoparticles in hyperthermia therapy, where these systems are exposed to alternating magnetic fields (typically ranging from ~100–300 kHz) and generate heat, mostly via magnetic hysteresis loss [22,126]. Due to hyperthermia, the temperature of cancer tissues might be raised up to 41–46 °C, triggering various paths as necrosis, apoptosis, protein denaturation, and immune system reactions [129]. Still, a is to ensure true tumoral tissue specificity, without damaging surrounding healthy tissues. Currently, several iron oxide nanoparticles have been approved for use in hyperthermia-based cancer therapy, such as NanoTherm^®^ and ThermoDox^®^ [126].

It has been described that that this class of nanoparticles can stimulate proinflammatory immune cell phenotypes, facilitating the recognition of tumors to enhance cancer therapies [126,130]. For example, Korangath et al. [74] recently reported the coupling of amine-functionalized starch-coated ferrite nanoparticles with a monoclonal antibody (HER2/neu), which has been clinically approved in therapies for breast cancer. After exposing cancer cells to these nanoparticles, the authors observed an infiltration of T cell populations (part of the immune system) into tumors, followed by tumor growth suppression. Similarly, the exposure of cancer cells to ferumoxytol [131], an example of an FDA-approved iron oxide nanoparticle, triggers an inflammatory response that leads to the prevention of metastases.

The functionalization of iron oxide nanoparticles with targeting ligands, antibodies, or peptides might enhance their selectivity towards cancer cell receptors or markers, facilitating targeted delivery of drug molecules (as doxorubicin and paclitaxel) [23,76] and short ribonucleotides (e.g., miRNAs, siRNAs) [132,133].

### 5.6. Carbon Nanomaterials

Carbon nanostructures are an important class of materials in the field of cancer therapies, including, for example, graphene-based structures, carbon nanotubes (CNTs), fullerenes, and carbon dots (CDs) [134], as illustrated in Figure 11. These carbonaceous structures have found applications as drug carriers and photoactive and diagnostic agents in several cancer theranostics [134,135]. A significant advantage of these systems lies in their large surface-area-to-volume ratios, which allows for enhanced loading and delivery of anticancer drugs towards the target cells, thus minimizing off-target effects. Moreover, because of their easy functionalization possibilities, the surface of these nanomaterials can be tailored to achieve different types of interactions (covalent and/or non-covalent) with drug molecules and ensure their controlled release to tumor sites [134].

The two-dimensional nature of graphene-based nanomaterials and their *sp*^2^ hybridization endow them with a unique honeycomb lattice structure to act as nanovehicles of anticancer drugs [136,137]. Within this context, oxidized derivatives of graphene, such as graphene oxide (GO), play a key role due to their higher dispersibility in physiological media and ability for chemical functionalization [138,139]. For example, Zhang et al. [140] loaded doxorubicin and camptothecin (CPT) onto GO to simultaneously explore the cytotoxic effect arising from DNA intercalation and topoisomerase inhibition in MCF-7 breast cancer cells. Moreover, due to their strong absorbance in the NIR region, it has been reported [141] that graphene derivatives can be stimulated by light to produce hyperthermia [142]. Additionally, these nanomaterials can aid typical photodynamic therapy due to their ability to carry multiple PSs that generate reactive oxygen species (ROS) under light irradiation.

CNTs assume special relevance in cancer treatment and diagnosis, namely when chemically functionalized with biocompatible molecules that increase their inner stability in physiological media. For example, Oh et al. [143] developed a delivery system that carried doxorubicin with PEGylated single-wall CNTs (SWNTs), which showed potential in chemotherapy and in combined NIR-irradiated PTT against human breast cancer cells [144]. Wen et al. [77] followed a similar rationale to load another anticancer drug (Sor) and EGFR onto multiwall CNTs (MWNTs): the results showed that this nanocomposite could decrease tumor growth in liver cancer cells, mostly by apoptosis. While attempting to target mitochondria, Yoong et al. [145] functionalized multiwall CNTs (MWNTs) with fluorescent rhodamine molecules to encapsulate a chemo-potentiator 3-bromopyruvate (BP) and platinum prodrug; the as-developed system led to mitochondrial malfunction, causing apoptosis of cancer cells.

The unique geometry and molecular topology of fullerene C_60_ consists of a round cage-type structure bearing 60 carbon atoms arranged in 12 pentagons and 20 hexagons [146]. Other fullerenes exist with other numbers of C atoms arranged in fused rings of five to seven atoms or with the surfaces functionalized with a variety of chemical groups. The abundant π–π conjugation of these nanomaterials endows them with important optical and thermodynamic properties, suitable for use as a photosensitizing agent in PDT, hyperthermia, imaging, and photoacoustic-assisted theranostics [147].

As a more recent member of this carbonaceous nanomaterial family, fluorescent CDs have been acquiring increasing importance in cancer therapies, namely in bioimaging [21,78]. Targeted staining of specific cancer cells using CDs typically relies on the attachment of special ligands, such as transferrin, folic acid, and hyaluronic acid [148,149,150]. These materials can also be used as delivery systems [151,152].

Despite these encouraging breakthroughs, the biocompatibility of carbon nanomaterials remains challenging: surface functionalization, modification, and encapsulation strategies have been employed to enhance biocompatibility, biodegradability, and control immune responses [153]. For example, CNTs have raised nanotoxicological concerns which prompt the necessity of more studies, namely associated with surface functionalization and biological impact. Hence, rigorous preclinical and clinical studies are still required to evaluate the safety and efficacy of carbon-nanomaterial-based cancer therapies.

## 6. Passive and Active Targeting

### 6.1. Passive Targeting

Nanocarrier-based cancer therapies are mostly passively targeted first-generation nanomedicines. This generation relies on manipulating pharmacokinetics and biodistribution by regulating physicochemical properties [154]. The pathophysiological properties of cancer and its environment have been used for inactive targeting, especially where the accumulation of nanomedicine in tumor cells is promoted by the EPR effect. Thus, nanomedicine treatments from passive targeting into neoplasms can occur by diffusion and convection without the attachment of a special substance to the nanocarrier surface.

Several studies have reported the use of liposome-based systems for passive targeting of drugs in cancer therapies. For example, Hamishehkar et al. [155] reported that sclareol functionalized with solid liquid nanoparticles (sclareol-SLNs, ~88 nm) has shown significant growth inhibitor effect on A549 human lung epithelial cancer cells after a period of 48 h, in comparison to the effect observed from free drug throughout a sustained drug release. Other examples involve the use of curcumin with SLN to target breast cancer [156] or Hodgkin’s lymphoma [157]. Regarding glioblastoma and melanoma, temozolomide-SLN was able to induce higher inhibition of proliferation and lower cytotoxicity when compared to non-functionalized temozolomide [158]. Examples that are clinically tested and used in the EPR effect are Doxil©, a liposomal delivery system of doxorubicin, and Abraxane©, an albumin-based nanoparticle delivery system of paclitaxel and Genoxol-PM© [159,160,161].

In spite of that, it is known that EPR-effect-based passive targeting is inefficient to control cytotoxic drug side effects. The drug delivery can be negatively affected through passive targeting due to the cancer heterogeneity and its stroma, and the consequence is a reduced or a nulled transport of the components into neoplasms [162]. This is the reason why researchers preferably focus on the standardization of neoplasm vasculature before starting cancer treatment Also, the extracellular matrix restricts drug penetration [163], and the accumulation of nanocarriers in former organs is not avoided by passive targeting [164]. Therefore, a next generation based on drug delivery with active directing transmitter nanocarriers having stimuli-reactive properties was developed, resulting in improved directing and enhanced efficiency potential [165] (Figure 11).

### 6.2. Active Targeting

According to Bazak et al., a high-affinity material annexes to the carrier surface area so the ligand can selectively bind to the target cell receptor [26]. Many ligand ranges (like carbohydrates and folic acid or macromolecules, like proteins, oligonucleotides, and aptamers) have been used with this intention. The preferred ligand binds to a targeted cell while it minimizes binding to healthy cells [13] (Figure 12).

For example, nanostructured lipid carriers (NLCs) coated with hyaluronic acid were used to load and deliver paclitaxel to cancer cells with overexpression on CD44 (a membrane glycoprotein), surpassing the effect of the free drug Taxol^®^ [166]. In another example [167], resveratrol-SLN was modified with apolipoprotein E, which enhanced its permeability through the BBB in comparison to the non-modified resveratrol-SLN. Cell-penetrating peptides can also be modified with SLNs to improve the antitumor efficiency: for example, Liu et al. [168] developed SLNs functionalized with trans-activating transcriptional activator (TAT) peptide, which contained two anticancer agents (tocopherol–succinate–cisplatin prodrug and paclitaxel).

## 7. Cancer Theranostics

Theranostics is a term used to describe systems that can diagnose, provide target drug delivery, and track the effects of the treatment. The main objective of bringing all these aspects together is to improve the chances of cure while minimizing the risks and the costs [40]. Thus, the intention behind designing this kind of agent is to develop a nano-sized system with a two-fold function. It is necessary to consider all the steps of this process, such as the method chosen or the materials that will be used in the preparation of the particles, and their removal from the body [169].

Different challenges have been acknowledged for the purpose of reaching an effective theranostic system. Zhao et al. and Bae et al. highlighted that two main characteristics must be considered in the biomarker that is used for imaging and treatment: it must be exceedingly expressed in the cancerous cells and absent in healthy cells [41,170]. Furthermore, it is required that the ligand must be highly reproducible for in vivo testing purposes, and it is crucial that the nanoparticles used in the theranostics agents are biocompatible, biodegradable, show a high loading capacity, and, when they reach the cancerous site, present a controlled release profile of the therapeutic elements [40].

The mentioned specifications are critical for the therapeutic part of a theranostic system, but the diagnosis part mainly represents imaging requirements. The nanoparticles should be capable of producing a constant and clear imaging signal in view to effectively monitor the targeted drug delivery as well as the response [171,172].

The abovementioned classes of nanoparticles have been employed in different theranostic strategies. Within this context, a special focus is given to magnetic-responsive theranostics, gene theranostics, radiation-responsive theranostics, and light-responsive theranostics, as comprehensively reviewed elsewhere [173].

The first one directly concerns MRI, which is an important imaging technology that provides non-invasive and accurate information about soft tissues, without the need for ionizing radiation or radiotracers. It has been reported that the accuracy of MR signals can be improved by using contrast agents containing several nanoparticles [174], particularly iron oxide nanoparticles, which have shown potential for early diagnosis of swelling and infection [175,176]. In another striking example, Gd-based liposomal MRI contrast agents have been used for targeted image-guided drug delivery.

Gene theranostics involves the infection of tumor cells with plasmid DNA (pDNA), which contains a gene that triggers tumor genetic defects and leads to the synthesis of a protein that promotes tumor cell death [177]. Within this framework, several synthetic and inorganic nanoparticles have been developed to act as gene vectors and to allow the pDNA to be delivered into tumor cells. For example, MRI-visible polymeric nanoparticles carrying cell-targeting pullulans were used for gene delivery into liver cancer cells [178]. In spite of its great potential, this strategy has not yet been approved in clinical procedures.

Radiation therapy (RT) relies on the use of minimal doses of X-rays to monitor the inside of the body. Metallic nanoparticles, whose photoelectric absorbance is high, can enhance the sensitivity of RT and enhance the radiation dose [179].

Light-mediated theranostics, including photothermal therapy (PTT) and photodynamic therapy (PDF), are among the most promising cancer therapies, already tested in vitro and in vivo. In particular, the use of near-infrared (NIR) light-mediated strategies (700–1000 nm) allows for a deep tissue penetration, minimum photodamage and low autofluorecence [180]. In one hand, PTT takes advantage of light energy to promote a temperature increase in target cells, leading to apoptosis via multiple pathways [181]. Several nanomaterials have been used for PTT, including gold nanoparticles [182], semiconductors [183], and other inorganic nanoparticles [184]. On the other hand, PDT relies on the exposure of photosensitizers (PSs) to light, which transfer the stored energy to nearby oxygen molecules, resulting in radical oxygen species (ROS) that lead to cell death [185].

Still, the implementation of theranostic nanoplatforms in a clinical setting faces significant challenges, such as the scale-up of their synthesis, the incoroporation within one nanoformula of imaging and therapy components, regulatory hurdles, and nanotoxicity studies [186]. Even so, it is undeniable that the integration of cancer theranostic nanosystems will revolutionize future healthcare and lead to a further step in the eradication of cancer. 

## 8. Conclusions and Future Perspectives

Nanomedicines brought a new perspective to cancer therapy mainly because of unique properties explored at the nanometric scale and the high bioavailability at the site of action. In this concise review, we have highlighted the most relevant novel outcomes concerning the use of nanoparticle-based approaches for cancer treatments throughout the last decade. In particular, this review has focused on the use of nanoliposomes, nanoemulsions, polymeric nanoparticles, quantum dots, gold nanoparticles, iron oxide nanoparticles, and carbon nanomaterials. Some of these structures have already found application in vivo, in vitro, and in clinical translation, due to the relevant features for application in cancer therapies, like size- and surface-dependent properties, versatility in their synthesis and surface modification, diverse functionalities, and modification to improve the required biocompatibility.

However, it is worth mentioning that in spite of the great improvements in the field, the use of nanotechnology in cancer therapy still faces many challenges ahead, which remain out of the scope of this review. Further studies focusing on their potential long-term effects in different biological systems are necessary to ensure their safe and scalable use as nanomedicines. Still, it is expected that much progress will be achieved in the near future not only in cancer treatments but also in other fields of medicine. In the ever-increasing body of literature, this concise review aims to inspire collaborative efforts, spark innovative solutions, and drive the field forward.

## Figures and Tables

**Figure 1 nanomaterials-13-02887-f001:**
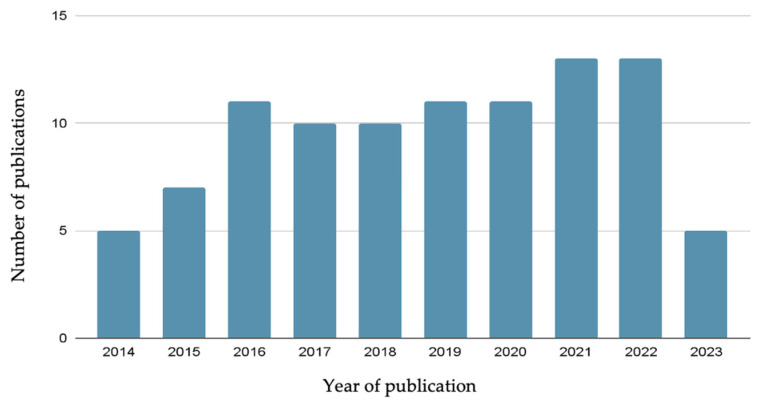
Correlation between the number of publications and year of publication of the research covered in this review.

**Figure 2 nanomaterials-13-02887-f002:**
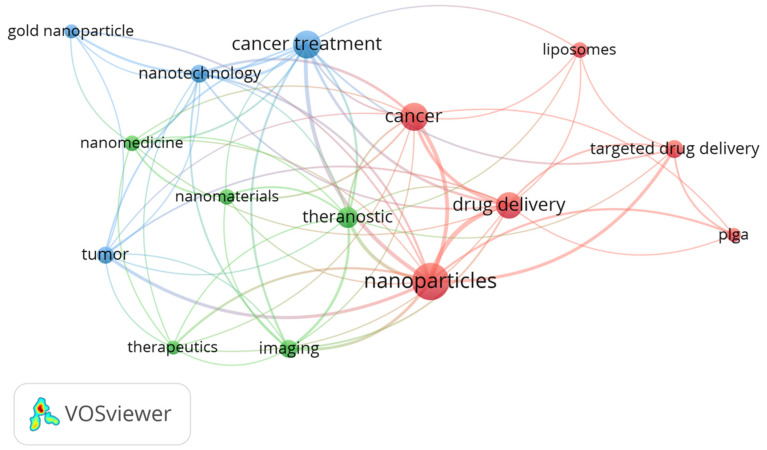
Map of the keywords found in the literature sources used in this review.

**Figure 3 nanomaterials-13-02887-f003:**
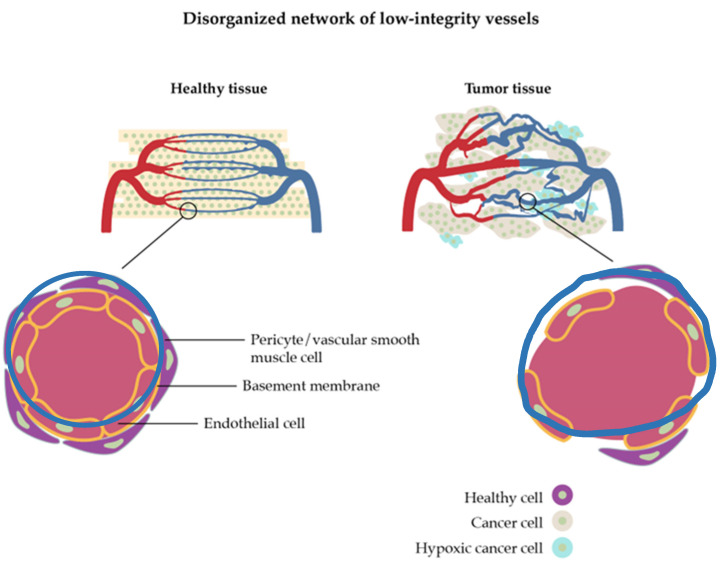
Schematic differences between healthy and tumor vasculature. Adapted from [39], with permission from Springer Nature, 2018.

**Figure 4 nanomaterials-13-02887-f004:**
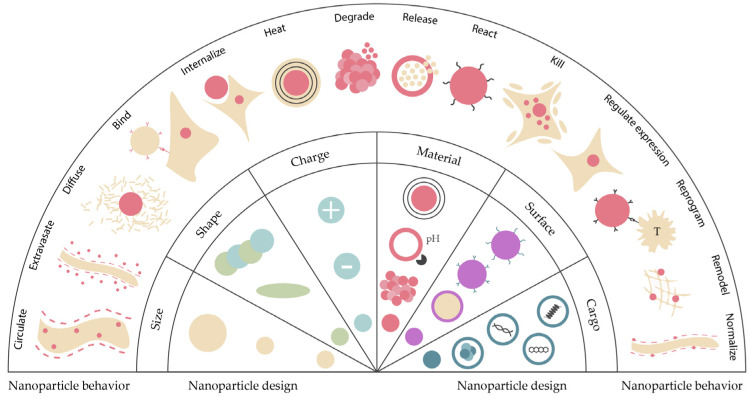
Physicochemical properties of nanoparticles, their various interactions inside the body, and their behavior inside the cell. Adapted from [45], with permission from Elsevier, 2014.

**Figure 5 nanomaterials-13-02887-f005:**
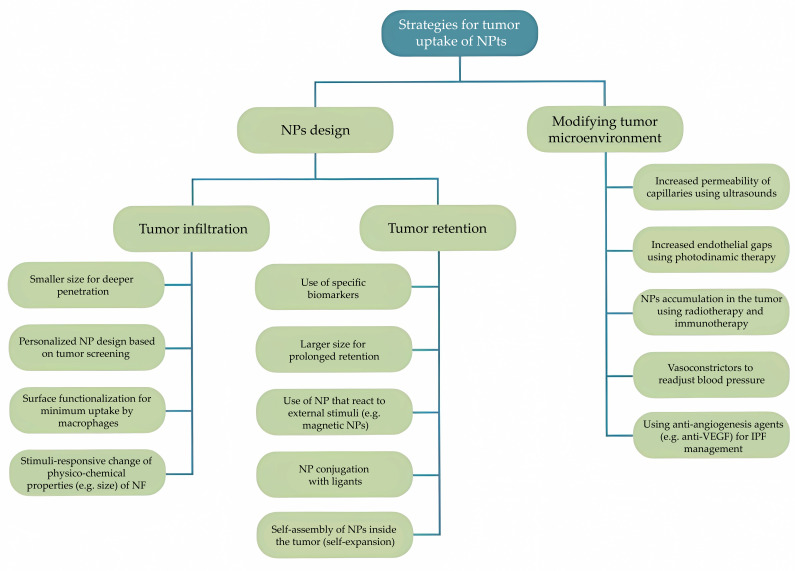
Different strategies for tumor uptake of nanoparticles. Adapted from [40], with permission from MDPI, 2019.

**Figure 6 nanomaterials-13-02887-f006:**
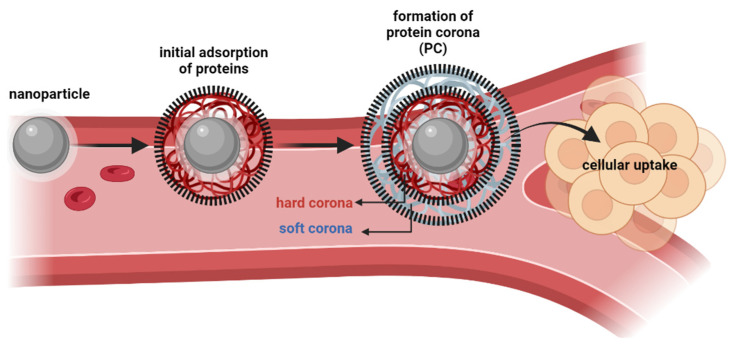
Formation of protein corona layers: hard corona and soft corona.

**Figure 7 nanomaterials-13-02887-f007:**
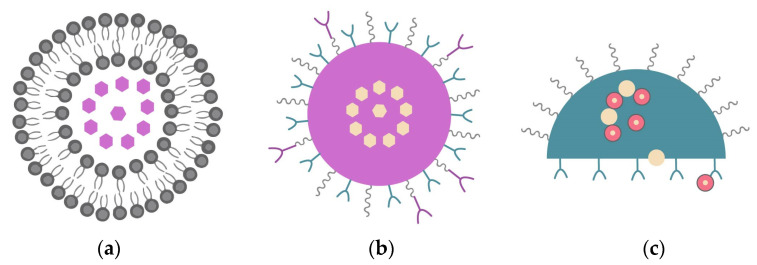
Schematic representation of three generations of nanovectors (**a**) first generation, (**b**) second generation, and (**c**) third generation. Adapted from [10], with permission from Elsevier, 2022.

**Figure 8 nanomaterials-13-02887-f008:**
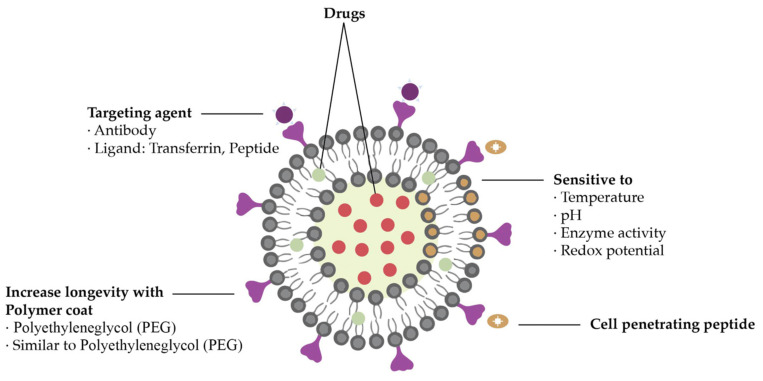
Liposomal drug delivery system. Adapted from [10], with permission from Elsevier, 2022.

**Figure 9 nanomaterials-13-02887-f009:**
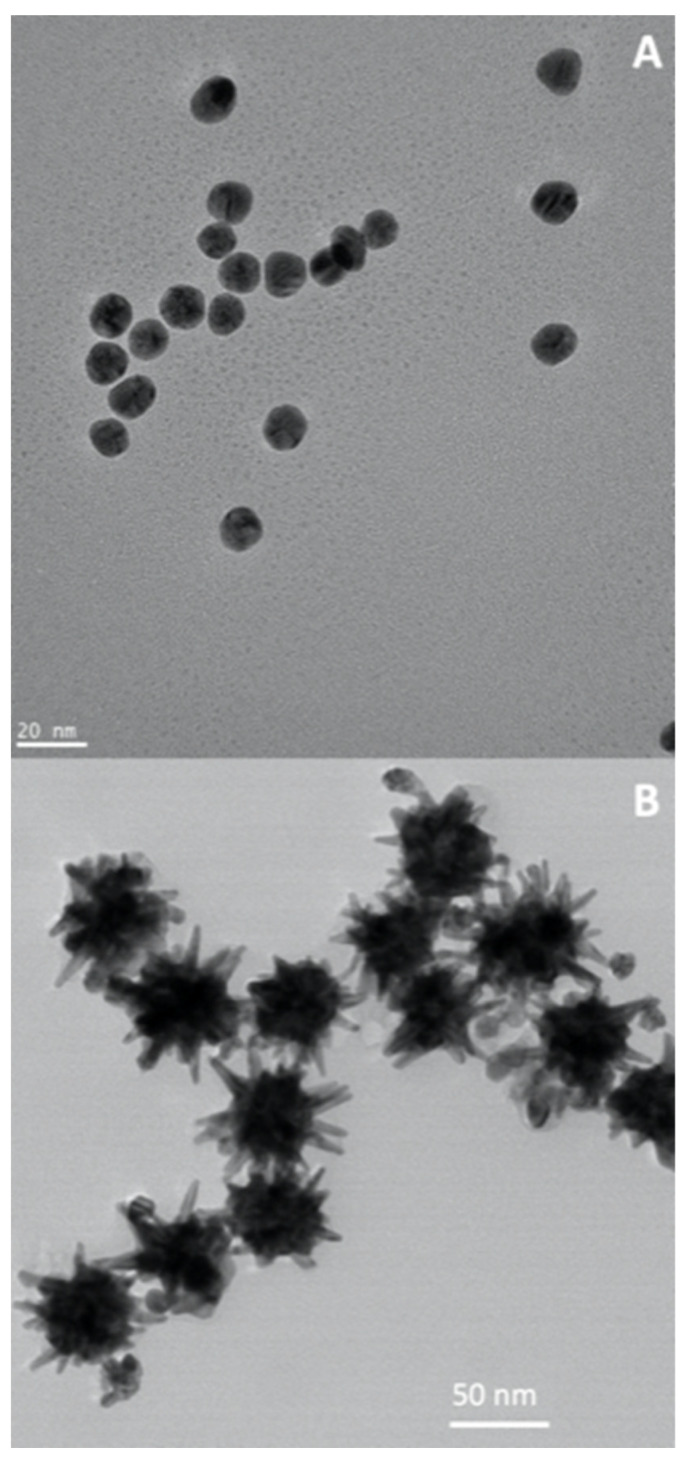
TEM images of Au nanospheres (**A**) and Au nanostars (**B**), both obtained by colloid synthesis. Courtesy of Dr. Sara Fateixa (U. Aveiro).

**Figure 10 nanomaterials-13-02887-f010:**
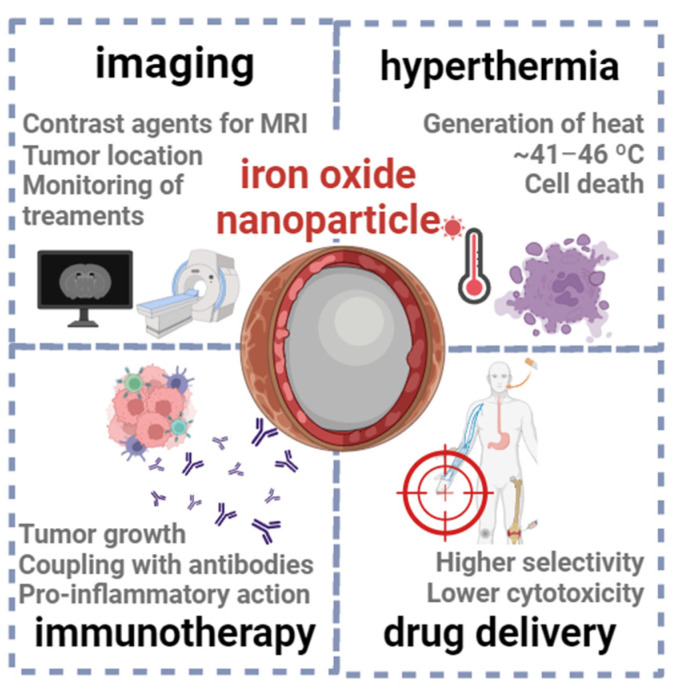
Examples of applications of iron oxide nanoparticles in cancer therapies.

**Figure 11 nanomaterials-13-02887-f011:**
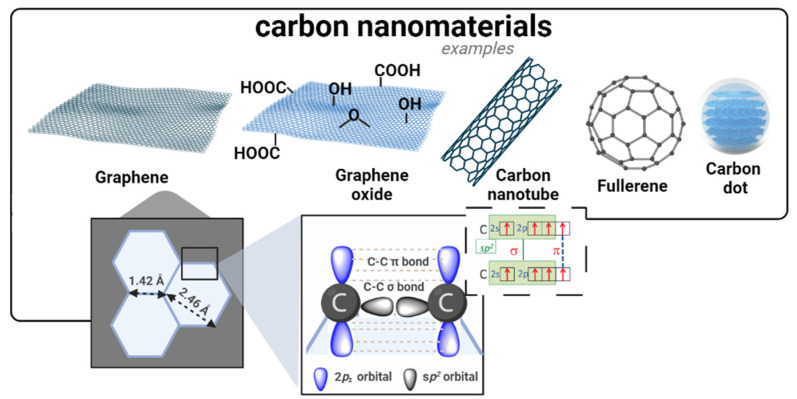
Examples of carbon nanomaterials. The structure and crystal lattice of graphene are shown.

**Figure 12 nanomaterials-13-02887-f012:**
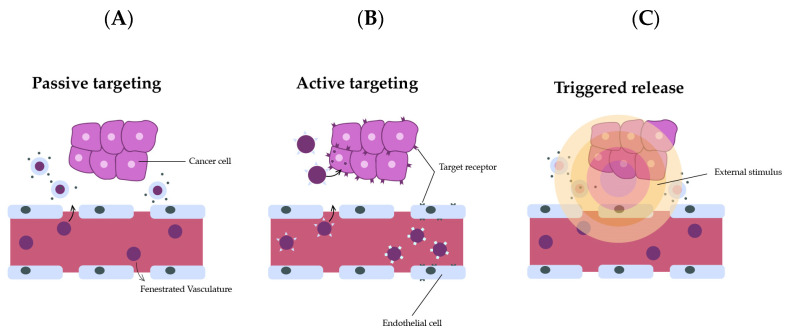
Types of targeting for nanoparticle delivery to tumor tissue. (**A**). Passive targeting relies on the leaky vasculature that is exhibited by tumors, allowing nanoparticles to travel through the fenestrations and reach tumors. (**B**). Active targeting can be used when nanoparticles have ligands on their surface that can recognize and bind receptors that are overexpressed on tumor cells. (**C**). Triggered release allows nanoparticles to congregate if exposed to an external stimulus such as a magnetic field or light. Adapted from [52], with permission from Springer Nature, 2017.

**Table 1 nanomaterials-13-02887-t001:** Clusters of the keywords in the sources used for this review article.

Cluster	Keywords	Weight (Links)	Weight (Total Link Strength)	Weight (Occurrences)	Score (Avg. Pub. Year)
1	Cancer	12	31	19	2013
	Drug delivery	12	37	17	2012
	Liposomes	5	5	6	2014
	Nanoparticles	13	69	34	2013
	PLGA	4	7	5	2010
	Targeted drug delivery	6	14	8	2011
2	Imaging	9	20	8	2015
	Nanomaterials	6	9	6	2019
	Nanomedicine	9	12	6	2019
	Theranostic	12	25	11	2018
	Therapeutics	8	10	5	2015
3	Cancer treatment	12	36	19	2015
	Gold nanoparticle	5	8	5	2018
	Nanotechnology	10	21	8	2017
	Tumor	9	16	7	2017

**Table 2 nanomaterials-13-02887-t002:** Examples of key nanoparticles used for cancer applications.

Type of nanoparticle	Formulation	Application	Ref.
Nanoliposomes	Liposomes and peptides/doxorubicin Liposomes and paclitaxel/carboplatin	Glioma tumor cells Enhanced MRI (in vitro, in vivo)	[58] [59]
Nanoemulsions	Nanoemulsion with fisetin Nanoemulsion with lycopene Nanoemulsion with photosensitizer (e.g., hexylaminolevulinate, aminolevulinic acid)	Lewis lung carcinoma (in vivo) Colon cancer (in vitro) Clinical trials	[67] [68] [69]
Polymeric nanoparticles	PLGA–PEG with paclitaxel PLGA–folic acid–chitosan with bicalutamide	Endometrial carcinoma (in vivo) Prostate cancer (in vitro)	[70] [71]
Quantum dots	Streptavidin-coated QDs (CdSe/ZnS) CdSe/CdS/ZnS QDs coupled to folic acid	Enhanced cytosolic delivery (in vitro) Mouth epidermal carcinoma (in vitro, in vivo)	[72] [73]
Gold nanoparticles	BSA-modified gold nanoparticles	Photothermal therapy (in vitro)	[19]
Iron oxide nanoparticles	Amine-functionalized starch-coated ferrite nanoparticles with monoclonal antibodies Ferumoxsil, Lumirem^®^ or Gastro MARK^®^	Breast cancer (clinically approved) MRI scans	[74] [75]
Carbon nanomaterials	GO with doxorubicin/camptothecin Multiwalled CNTs with sorafenib CDs	Breast cancer (in vitro) Liver cancer (in vitro) Bioimaging	[76] [77] [21,78]

## Data Availability

Data sharing not applicable.

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
