# Peer review of "Application of Nanoparticles in Cancer Treatment: A Concise Review"

_nanomaterials, 2023, doi:10.3390/nano13212887_

Round 1

Reviewer 1 Report

Comments and Suggestions for Authors

Luisa Cruz-Lopes and team reviewed a topic entitled “Application of nanoparticles in cancer treatment: a concise review.” The authors mainly focused on the different types of nanoparticles, ranging from inorganic and organic nanoparticles used in cancer diagnosis, imaging, and treatment. The authors also discussed nanoparticle action mechanisms, benefits, and applications. The team made an effort, to sum up the structural parameters of the nanoparticles necessary for effective cancer diagnosis and therapy from the biological perspective.

This manuscript is worth publishing in the MDPI Biomolecules. However, we recommend a detailed revision addressing the following issues carefully to reach more audiences and readers of different disciplines before considering a possible publication. 

1. Some acronyms or abbreviations are unnecessary if the information was mentioned only once in the manuscript. They will just increase the word count without specific scientific information. 

For example,

Page 3, Line 109: (MDR/ATP)

Page 7, Line 213: (HPLC)

Page 11, Line 377: (VIS)

Page 12, Line 394: (SERS)

Page 13, Line 455: (CT)

2. Expand the full form of acronyms or abbreviations. 

Page 11, Line 361: ZnS  Zinc sulfide

Page 11, Line 361: InP  Indium phosphide

3. Some sentences are confusing because of English.

Page 10, Line 326: Polysorbate 80 is known to enhance the nanoparticles loaded blood-brain barrier (BBB) cross. Polysorbate 80 is known to enhance the transport of nanoparticles across the blood-brain barrier (BBB). Citations were missing for this statement (J Drug Target 2006;14(2):97-105 and Eur J Neurosci 2000;12(6):1931-40). 

4. Page 5, Figure 2: Hypoxic cells were not shown in the magnified image of Figure 2.

5. Some important scientific discoveries were not cited properly. For example, a polymeric micelle of 30 nm size showed fast transcytosis and, thus, efficient extravasation and infiltration into poorly permeable tumors, thereby showing enhanced antitumor activity compared to micelle size of 70- and 140-nm size (Nat Nanotechnol 2011;6(12):815-23 and Nano Lett 2023;23(9):3904-3912).

6. A single hurdle to the clinical translation of nanomedicines is the accumulation of the majority of the administered dose of nanomedicines by the reticuloendothelial system (RES) organs, mainly the liver. Hence, we recommend the authors discuss this important information and show the strategies that help overcome the RES barrier with the suggested pieces of information. Other strategies are also welcome.  

Reticuloendothelial (RE) capture of nanoparticles is one of the critical hurdles to the translation to the clinics because these RE cells not only capture the majority of the injected dose, thereby drastically limiting the dose of nanoparticles to the disease area, but also raise immunogenicity and toxicity issues (Adv Drug Deliv Rev 2023;198:114895 and Nat Mater 2016;15(11):1212-1221). Temporary blocking of the RE cells dramatically improved the drug delivery efficiency to the disease sites by rerouting the nanoparticles from the RE organs without the need for special ligands (Nat Commun 2023;14(1):1437, Nat Biomed Eng 2020;4(7):717-731, Sci Adv 2020;6(26):eabb8133, and ACS Nano 2023;17(10):8966-8979). 

7. PEGylation is a widely used strategy for extending blood circulation, thereby improving therapeutic outcomes in vivo. However, PEGylation compromises the uptake and endosomal escape efficiency (termed PEG dilemma). To overcome this dilemma, several strategies were introduced onto the surface of nanoparticles to improve cancer treatment and diagnosis. We recommend the authors discuss the possible strategies to overcome this PEG dilemma.  

(a) A polyion complex micelle was developed by self-assembling ethylenediamine-based polycarboxybetaine polymers with pDNA (J Control Release 2023;360:928-939). This micelle switched its surface charge to a positive charge in response to tumorous (pH 6.5) and endolysosomal acidic milieu (pH 5.5) from its original neutral charge at pH 7.4 (bloodstream), thereby promoting the cellular uptake and endosomal escape toward efficient gene transfection. The cargo pDNA of this micelle encodes a soluble form of soluble fms-like tyrosine kinase-1, a potent antiangiogenic exogenous protein, which captures vascular endothelial growth factor (VEGF), thereby significantly suppressing the growth of hard-to-treat solid tumors. 

(b) Tumor neovasculature endothelial cell targeting ligands (for example, cyclic Arg-Gly-Asp) are strategically appended to the distal end of the PEG shell for promoting tumor cell uptake of nanoparticles via specific integrin-mediated uptake(Biomaterials 2014;35(20):5359-5368). 

8. Recent results challenge the transport of nanoparticles through inter-endothelial gaps of the tumor blood vessels, which is a central paradigm in cancer nanomedicine. We also recommend the team introduce such alternative active transcytosis pathways (Nat Mater 2020;19(5):566-575 and Adv Drug Deliv Rev 2022:189:114480).

Comments on the Quality of English Language

Some sentences are confusing because of English. Page 10, Line 326: Polysorbate 80 is known to enhance the nanoparticles loaded blood-brain barrier (BBB) cross. Polysorbate 80 is known to enhance the transport of nanoparticles across the blood-brain barrier (BBB)

Author Response

I would like to express my gratitude for your comments on my article. Your observations and suggestions will serve to further strengthen my work.

Reviewer 2 Report

Comments and Suggestions for Authors

Sell et al. have made a concise review on application of nanoparticles in cancer treatment. The topic is too wide and deep. Although the topic is of interest, this article requires substantial improvement based on the following points for a possible publication in Nanomaterials:

1.        The abstract should be improved to cover the elucidating points from the discussion made in this review article.

2.        The type of nanoparticles covered in this review should be mentioned in the abstract.

3.        Some more keywords such as ‘tumor environment’, ‘passive and active targeting’, ‘nanomedicine’ should be added.

4.        The year range of published paper the authors used for review should be mentioned in the abstract and section 2.

5.        Under the section 2, in addition to the provided bibliometric analysis, publication growth data obtained from ‘Web of Science’ should be included by plotting no. of publications vs year (last 10 years) to highlight the growing importance of this topic.

6.        In all the figure captions, it should be ‘adapted with permission from [ ]’.

7.        The main drawback in the discussion is the lack of any tables to summarize the key data/points from the published papers used for discussion. Therefore, it is important to prepare tables on in vitro, in vivo and clinical applications of the discussed nanoparticles.

8.        Why in the list of nanoparticles, ‘Nanoemulsion’ is left out? If it is said that it is not a ‘nanoparticle’, then ‘liposomes’ are also not nanoparticles.

9.        The section 6 should be expanded with much more content involving key articles with specific nanoparticles used for passive targeting and active targeting. Otherwise, the information is too little, but the importance of it is huge.

10.     The conclusions section should be modified to be ‘conclusions & future perspectives’ with the contents including elucidations from the discussions on subtopics covered, limitations of this review, and research gaps/future perspectives identified based on this review should be all be included in this section.

Comments on the Quality of English Language

Minor editing of English language required

Author Response

(The authors gave the same response as above.)

Reviewer 3 Report

Comments and Suggestions for Authors

nanomaterials-2665589

Application of nanoparticles in cancer treatment: a concise review

The review manuscript by Sell et al. summarized different types of nanoparticles used in cancer therapy, their action mechanisms, and their benefits and applications in diagnosis, imaging, and treatment. This manuscript was well structured. The authors presented several aspects of the nanoparticles in cancer treatment. However, there are some concerns in this manuscript as follows.

1. Various similar reviews are available. It is hard to see the novelty and contribution of this review. What makes this review different from the other reviews?

2. The topic “Nanoparticles in cancer treatment” is too broad. As can be seen, the content presented in this review is relatively superficial. The Conclusion is too general.

3. Section 2: The authors should follow PRISMA guidelines to search the articles and present the results of the article search and screening processes. The authors must clarify which databases were used for the search.

4. Figures adapted from previous articles should come along with a statement of copyright and reuse permission. The permission (if necessary) should be included as a Non-published SI.

Comments on the Quality of English Language

Minor editing of English language required

Author Response

(The authors gave the same response as above.)

Round 2

Reviewer 1 Report

Comments and Suggestions for Authors

Accepeted in the present form 

Author Response

I would like to express my gratitude for your contribution on my article.

Reviewer 2 Report

Comments and Suggestions for Authors

The authors have satisfactorily addressed all the comments raised by reviewers and therefore I recommend acceptance of this article for publication in Nanomaterials.

Comments on the Quality of English Language

Minor editing of English language required

Author Response

(The authors gave the same response as above.)

Reviewer 3 Report

Comments and Suggestions for Authors

nanomaterials-2665589

Application of nanoparticles in cancer treatment: a concise review

The manuscript was revised accordingly and can be accepted after a major revision.

1. Section 7 should be expanded to discuss in-depth cancer theranostics.

2. Please discuss the clinical state of nanoparticles in cancer treatment.

3. Please check and correct typos and grammar errors (e.g, "theragnostics").

Comments on the Quality of English Language

Minor editing of English language required

Author Response

(The authors gave the same response as above.)

Round 3

Reviewer 3 Report

Comments and Suggestions for Authors

The revised manuscript can be accepted as is.